# Punicalagin in Cancer Prevention—Via Signaling Pathways Targeting

**DOI:** 10.3390/nu13082733

**Published:** 2021-08-09

**Authors:** Izabela Berdowska, Małgorzata Matusiewicz, Izabela Fecka

**Affiliations:** 1Department of Medical Biochemistry, Wroclaw Medical University, Chałubińskiego 10, 50-368 Wroclaw, Poland; 2Department of Pharmacognosy and Herbal Medicines, Wroclaw Medical University, Borowska 211A, 50-556 Wroclaw, Poland; izabela.fecka@umed.wroc.pl

**Keywords:** punicalagin, pomegranate, ellagitannins, breast cancer, cervical cancer, ovarian cancer, colorectal cancer, thyroid cancer, apoptosis, autophagy

## Abstract

The extract of pomegranate (*Punica granatum*) has been applied in medicine since ancient times due to its broad-spectrum health-beneficial properties. It is a rich source of hydrolyzable tannins and anthocyanins, exhibiting strong antioxidative, anti-inflammatory, and antineoplastic properties. Anticancer activities of pomegranate with reference to modulated signaling pathways in various cancer diseases have been recently reviewed. However, less is known about punicalagin (Pug), a prevailing compound in pomegranate, seemingly responsible for its most beneficial properties. In this review, the newest data derived from recent scientific reports addressing Pug impact on neoplastic cells are summarized and discussed. Its attenuating effect on signaling circuits promoting cancer growth and invasion is depicted. The Pug-induced redirection of signal-transduction pathways from survival and proliferation into cell-cycle arrest, apoptosis, senescence, and autophagy (thus compromising neoplastic progression) is delineated. Considerations presented in this review are based mainly on data obtained from in vitro cell line models and concern the influence of Pug on human cervical, ovarian, breast, lung, thyroid, colorectal, central nervous system, bone, as well as other cancer types.

## 1. Pomegranate against Pathologies Development

Health profits of regular consumption of polyphenol-rich foods including fruits such as pomegranates, green tea or red wine, have been extensively studied in recent years [1,2,3,4,5,6,7,8]. Their valuable health effects have been attributed not only to the activity of dietary fiber fractions, vitamins, and anthocyanins, but also to ellagitannins, which after intake are converted into ellagic acid, urolithins, and other low-molecular-weight phenols. The results of available epidemiological studies confirm that a diet rich in plant polyphenols reduces the risk of such chronic disorders as obesity [3], associated with increased risk of metabolic syndrome and atherosclerosis, diabetes [1,2], cardiovascular diseases [4], as well as cancer [5,6]. One of the most thoroughly examined plants, with respect to its broad-spectrum health-beneficial properties, is pomegranate (*Punica granatum* L.), a fruit applied in medicine since ancient times [7]. The chemical composition of pomegranate, with respect to the distribution of its components among different parts of the plant, as well as specific varieties, has been recently reviewed by Pirzadeh et al. [8] and Ge et al. [7]. Pomegranate is a rich source of hydrolyzable tannins and anthocyaninns which exhibit strong antioxidative and anti-inflammatory properties. Hydrolyzable tannins (ellagitannins and gallotannins), being the major class among pomegranate phenolic compounds, seem to be the main substances responsible for its observed favorable characteristics, and the major compounds present within this group comprise punicalagin, punicalin, granatins A and B, tellimagrandin I, pedunculagin, corilagin, as well as their hydrolytic products, namely gallagic acid, ellagic acid, and esters of glucose [9]. The group of pomegranate anthocyanins includes delphinidin-3-glucoside, cyanidin-3-glucoside, delphinidin-3,5-diglucoside, cyanidin-3,5-diglucoside, pelargonidin-3,5-diglucoside, and pelargonidin-3-glucoside) [8,10,11]. Pomegranate’s health-beneficial properties have been lately addressed with respect to its antimicrobial functions [8]. Additionally, the scientific data elaborating pomegranate’s role in the prevention and therapy of inflammatory disorders [12], as well as cancer diseases [11,13], have been up-dated. Extracts derived from various *P. granatum* parts, as well as purified components have been demonstrated to interfere with multiple signaling pathways involved in the development of pathological conditions. Sharma et al. [11] summarized the literature data associated with molecular targets of pomegranate’s components whose administration attenuated tumor development through the induction of apoptosis, and/or inhibition of proliferation, cell-adhesion, metastasis, and neoangiogenesis.

## 2. Anti-Cancer Activities of Punicalagin (Pug)

Punicalagin (CAS No: 65995-63-3; (Figure 1)) belongs to ellagitannins—a subgroup of hydrolyzable tannins. It is abundant in pomegranate juice, fruits, peel (pericarp), seeds, flowers, leaves and bark, as well as in the fruits of myrobalan (*Terminalia chebula* Retz.), leaves of yellow wood (*Terminalia oblongata* F. Muell.), tropical almond (*Terminalia catappa* L.) [9,14] and pink rock-rose (*Cistus* × *incanus* L.) [15]. In the aqueous environment, it readily undergoes hydrolysis releasing ellagic acid (EA, CAS No: 476-66-4), gallagic acid dilactone (terminalin, CAS No: 155144-63-1) [16], and D-glucose.

Both punicalagin and its derivative—ellagic acid have been demonstrated to protect DNA from mutations [17]. Zahin et al. [17] observed that these two compounds dose-dependently diminished the level of DNA damage caused by a variety of carcinogens. For instance, almost complete inhibition of DNA adducts formation by Pug (40 µM) was noted in the experimental model with benzo[a]pyren, DNA, and rat liver microsomes (containing cytochrome P-450 associated biotransformation system required for the conversion of benzopyren into a mutagen). Additionally, the authors noted a dose dependent cytotoxic/antiproliferative effect of both Pug and EA on human lung cancer cell lines A549 and H1299 (MTT assay). Seeram et al. [18] demonstrated that pomegranate juice (PJ) and pomegranate tannin extract, both rich in punicalagin, inhibited the proliferation of human oral (KB, CAL27), colon (HT-29, HCT116, SW480, SW620), and prostate (RWPE-1, 22Rv1) tumor cells. Moreover, pure Pug as well as its hydrolytic product EA showed antiproliferative action on these cell lines. However, the effect of PJ was the strongest, suggesting the synergistic/additive accomplishment of other phytochemicals contained in PJ, as was hypothesized by the authors [18].

### 2.1. Punicalagin against Cervical Cancer

Punicalagin has been shown to exert antiproliferative/proapoptotic impact on human cervical cancer cell lines [19,20]. Zhang et al. [19] observed the cytotoxic effect of Pug on ME-180 cells in a dose-dependent mode (up to 100 µM). Similarly, Tang et al. [20] demonstrated a dose- (up to 200 µM) and time-dependent antiproliferative action of Pug on HeLa cells. Both scientific groups have noted the upregulation of proapoptotic Bax proteins and downregulation of antiapoptotic Bcl-2 factors resulting from Pug treatment. Additionally, Pug stimulated the expression of tumor suppressor p53 gene, as well as proapoptotic caspases 3 and 9 [19]. Moreover, the putative influence of Pug on signaling pathways, diverting cells into carcinogenesis/metastasis pathways has been investigated in those two studies. Tang et al. [20] demonstrated a dose-dependent inhibitory effect of Pug on β-catenin, C-myc, and cyclin D1 expression at the protein level. β-catenin/Wnt signaling pathway has been implicated in the progression of many cancers including cervical cancer, so its attenuation by Pug might be associated with cancer growth limitation. Furthermore, the authors observed the inhibition of HeLa cell migration in a wound-healing test, which was in agreement with the downregulation of matrix metalloproteinases 2 and 9 (MMP-2 and 9), and upregulation of their inhibitors, namely tissue inhibitors of metalloproteinases 2 and 3 (TIMP-2 and 3), resulting from Pug treatment. The interplay between proteolytic enzymes such as MMPs and their inhibitors, TIMPs, affects extracellular matrix (ECM) remodeling/degradation, and shifting of the balance towards increased proteolysis of ECM components exerted by MMPs is associated with the augmentation of cancer cells invasive capabilities. Therefore, Pug seems to be a promising agent aimed at compromising cervical cancer cells invasion and metastasis. Zhang et al. [19] found Pug to increase NF-κB-p65 protein expression in the cytosol but decrease its expression in the nucleus. Since the translocation of NF-κB to the nucleus is associated with its induction of target genes expression, their results suggest the attenuation of the signaling pathway associated with this component. NF-κB signaling route has been implicated in tumor initiation, progression and chemoresistance, including cervical cancer [21]. This so-called canonical pathway can be triggered by a variety of stimuli, such as proinflammatory cytokines (e.g., TNFα or IL-1), and is associated with the survival promotion. However, an alternative non-canonical mode of this pathway activation seems to function in a reversed manner rather facilitating apoptosis and thus suppressing tumor growth [21]. The direction of NF-κB action is context-dependent, related to multiple factors, such as cancer stage and type, as thoroughly discussed in the latest review publications [22,23,24], also with respect to its targeting by polyphenols [25]. Nevertheless, punicalagin effect on cervical cancer cells was evidently proapoptotic and antiproliferative, hence the canonical NF-κB route must have been silenced in the work of Zhang et al. [19].

The main findings on the impact of punicalagin on cervical cancer, as well as other cancer types, discussed in further paragraphs, are summarized in Table 1.

### 2.2. Punicalagin against Ovarian Cancer

Tang et al. [27] detected Pug effect against human ovarian cancer cells A2780 whose viability was dose- and time-dependently reduced, accompanied by cell cycle arrest and apoptosis induction. Similarly, as in their aforementioned studies on cervical cancer [20], the authors found Pug to compromise ovarian cancer growth through the inhibition of β-catenin signal-transduction pathway, as well as to reduce the migration of cells by attenuating matrix metalloproteinases and boosting their inhibitors. The involvement of hyperactivated Wnt/β-catenin signaling in gynecological malignancies, especially ovarian neoplasia, with respect to its silencing as a therapeutic aim, is addressed in Section 3.1.2.

### 2.3. Punicalagin against Breast Cancer

Pan et al. [26] analyzed the impact of Pug on the factors associated with the stimulation of cancer cells motility (migration and invasion) through the mechanism termed epithelial to mesenchymal transition (EMT), which proceeds via the upregulation of N-cadherin paralleled by the down-regulation of E-cadherin. Pug pretreatment of two human breast cancer cell lines (MCF-7 and MDA-MB-231) resulted in the reversed pattern of the two cadherins expressions: E-cadherin protein level rose, whereas N-cadherin dropped, being indicative of EMT inhibition. Additionally, Pug dampened the expression of metalloproteinases (MMP2 and 9) associated with the acquirement of the invasive and metastatic cancer phenotype. Furthermore, the authors observed the participation and Pug-modulation of Golgi phosphoprotein 3 (GOLPH3) in the promotion of invasive potential in the investigated breast cancer cells. Whereas GOLPH3 overexpression enhanced the cells’ motility and up-regulated the pro-invasive factors mentioned above, Pug attenuated these effects. Oncogenic aspects of GOLPH3 in many malignancies including breast cancer, and the consequences of its upregulation have been recently discussed by Sechi et al. [38]. GOLPH3 overexpression has been correlated with the promotion of neoplastic phenotype, worsening patients’ prognosis and heightening resistance to chemotherapy [39,40,41]. In normal cells, GOLPH3 is required for the proper functioning of the Golgi apparatus, whereas its disturbances such as abnormal vesicle trafficking contribute to the development of neoplastic phenotype. Overactivation of the GOLPH3/MYO18A pathway stimulates the exocytosis of growth factors and matrix metalloproteinases, enhancing cell invasive potential. GOLPH3 overexpression modifies integrin-mediated signaling, resulting in cytoskeleton reorganization and cell migration. GOLPH3 also responds to DNA damage leading to massive Golgi fragmentation which increases the neoplastic cell surveillance [38]. Therefore, as observed by Pan et al. [26], the GOLPH3-opposed Pug impact on breast cancer cells, paralleled by the reduction of EMT and invasive potential, seems promising with respect to anticancer Pug application.

### 2.4. Punicalagin against Colorectal Cancer

Ganesan et al. [31] observed proapoptotic impact of Pug on human colorectal cancer cell line HCT116, reflected by the increase in early apoptotic cells (assessed by Annexin V/PI flow cytometry) and cytochrome C release from the mitochondrium. This finding is in accordance with the results obtained by Seeram et al. [18] showing an antiapoptotic Pug effect in HCT116 as well as HT-29 (human colon adenocarcinoma) cell lines. However, Ganesan et al. [31] observed that 72-h Pug treatment of HCT116 cells downregulated caspases 3/7, 8, and 9. Hence, the authors checked for the features indicative of autophagy stimulation by Pug, and with the application of autophagy flux assay via flow cytometry, noted autophagosomes degradation (reflecting autophagy involvement). Additionally, they analyzed Pug effect on annexin A1 expression and noted a significant downregulation of this protein. Annexins are ubiquitously occurring calcium-dependent proteins which recognize and bind membranous phospholipids. Annexin A1 is involved in a variety of physiological processes, including differentiation, proliferation, as well as apoptosis. However, its best described properties are associated with the attenuation of inflammatory processes (e.g., through the inhibition of phospholipase A2 and therefore reduction of arachidonic acid release, the main precursor of proinflammatory eicosanoids) [42]. Annexin A1 faulty expression has been associated with cancer disease, but its actual role is unclear. As discussed by Fu et al. [42], annexin A1 function in malignancies might be seen as a two-edged sword due to the fact that, in some cancers, it seems to act as a tumor suppressor, whereas in others as a tumor promoter. For instance, in prostate [42,43] or oesophageal [44] cancers, a downregulation of this protein has been observed, whereas in hepatocellular [45] as well as colorectal cancers [46,47] annexin A1 elevated expression has been detected. Therefore, in colorectal cancer, this protein seems to favor neoplastic development, hence its suppressing by Pug is favorable. Further, Ganesan et al. [31] performed a proteome profiling analysis testing the impact of Pug (with annexin A1-route involvement) on 35 different proteins associated with apoptosis and autophagy processes. They observed four significantly altered proteins in HCT116 cells treated both with punicalagin and formyl peptide receptors (FPR) inhibitors (applied to block annexin A1 signaling). Three of them, i.e., heat shock proteins 27 and 60 (HSP27 and HSP60) and tumor necrosis factor receptor 1 (TNF RI), were down-regulated, whereas catalase was upregulated. Since the first two factors are associated with autophagy regulation (HSP27, HSP60), whereas TNF RI induction yields apoptotic events, the authors concluded that the stimulation of both pathways leading via annexin A1 suppression may be responsible for Pug-induced colorectal cells death in their research model. Adams et al. [32] investigated the effect of Pug on human colon cancer cell line HT-29 stimulated with TNF-α. The authors observed a dose-dependent reduction in COX-2 expression in the cells pretreated with increasing Pug concentrations (up to 200 mg/L). However, this effect was stronger after the application of pomegranate juice (PJ) or total pomegranate tannin extract (TPT). Moreover, PJ treatment was shown to attenuate TNF-α-induced signaling pathway leading via AKT and NF-κB. Upregulation of this type of proinflammatory pathway, stimulating the expression of COX-2 and other inflammatory factors such as cytokines, has been observed in various cancers including colorectal cancer. Therefore, the anti-inflammatory Pug effect observed by the authors is promising with respect to its preventive application in cancer diseases. However, the authors underlined that many beneficial effects observed in theirs and aforementioned studies [18] are enhanced when mixtures of tannins or pomegranate juice is applied, and not purified Pug. Larrosa et al. [33] noted the Pug-induced arrest of cell cycle at S phase, and its proapoptotic effect (via the intrinsic-mitochondrial pathway stimulation) on human colon cancer cell line Caco-2. Nevertheless, the authors’ profound analysis demonstrated that it was rather ellagic acid released from hydrolyzed Pug molecule which exerted all of these effects in their experimental model. Therefore, it might be hypothesized that many effects interpreted as an impact of punicalagin are the consequences of its degradation products.

### 2.5. Punicalagin against Thyroid Cancer

Cheng et al. [34] in their studies on human papillary thyroid carcinoma cell line (BCPAP) observed a reduction of the cell viability upon Pug treatment in a concentration and time-dependent mode (after 24-, 36-, and 48-h incubation with doses up to 100 μM). However, the closer analysis of this effect demonstrated apoptosis-independent mechanism associated rather with the induction of autophagy (type II programmed cell death). The authors did not detect characteristic features indicative of apoptotic events (e.g., neither the cleavage of caspase-3 nor poly(ADP-ribose) polymerase (PARP) were noted). Reversely, Pug stimulated-cancer cells expressed the profile of markers associated with autophagy; increase in autophagic vacuoles, increase in microtubule-associated protein light chain 3 II (LC3-II) conversion, beclin-1 expression, and p62 degradation. Additionally, the authors observed the stimulation of MAPK signaling pathway (increase in ERK 1/2 and p38 phosphorylation) and inhibition of mTOR route (drop in phosphorylated p70, S6, and 4E-BP1). These findings allowed them to conclude that the reversed mode of the two signal-transduction pathways modification exerted by Pug led to the final lethal effect via autophagy in the studied thyroid cancer cells. In the further studies of the same team, Yao et al. [35] analyzed the effect of Pug on DNA damage in BCPAP cells, as well as its modulation of DNA damage response (DDR) pathway. The authors observed that Pug-induced over 5-fold reduction in the cell viability was accompanied by the comparable degree of phosphorylated-H2A.X amplification. H2A.X is a form of 2A histone being a sensitive marker of DNA damage, and its phosphorylation indicates the initiation of DDR pathway. Moreover, 24-h treatment with 100 μM of Pug augmented the phosphorylation of ATM kinase (ataxia telangiectasia mutated) but did not affect ATR (ATM and rad3-related) kinase phosphorylation. Finally, the authors suggested that Pug contributed to the studied cells’ death via the induction of double strand breaks (DSB) in DNA helix, followed by the stimulation of DDR pathway associated with ATM signaling. In their ensuing experiments, the authors reported the Pug-induced acquirement of the senescent phenotype of the BCPAP thyroid cancer cells [36]. Pug altered the cells’ morphological features, increased their granularity, and upregulated both senescence-associated beta-galactosidase (SA-beta-Gal) as well as cyclin-dependent kinase inhibitor p21. The observed effects seem to have been mediated by NF-κB signaling pathway, which activation by Pug was found in this study. Namely, Pug was shown to induce the translocation of NF-κB-p65 to the nucleus. However, there is a discrepancy between the authors comment on Pug effect on IκBα and the results presented in Figure 4A [36] included in their paper. The authors concluded that Pug induced the phosphorylation and degradation of IκBα. However, judging from this figure, Pug seems to have stimulated only the total IκBα factor, and not its phosphorylated form [36].

### 2.6. Punicalagin against Lung Cancer

Fang et al. [28] observed punicalagin to exert concentration-dependent (up to 30 μM) pro-apoptotic effect on human lung cancer A549 cells. The authors demonstrated Pug-induction of proapoptotic factors (Bax; caspases 3 and 9; cytochrome C), and inhibition of anti-apoptotic Bcl-2. Similar results have been obtained by Berkoz et al. [29] in the same type of cancer cells which underwent apoptosis upon Pug treatment (dose-dependently at 50 and 75 μmolar levels), associated with the increase in caspases 3, 8, and 9 expression. Moreover, the induction of reactive oxygen species (ROS) by Pug was noted in both studies, however Berkoz et al. [29] observed ROS stimulation only in the mitochondria. Additionally, Pug was found to inhibit JAK/STAT signaling pathway in the lung cancer cells. Pug treatment reduced Jak-1 protein expression, and affected the STAT-3 protein expression profile, inducing its expression in the cytosolic fraction and inhibiting the expression of STAT-3 in the nucleus [28]. Therefore, as the authors suggested, Pug seemed to be able to inhibit this transcription factor translocation to the nucleus. Since Stat3 has been shown to inhibit apoptosis by the induction of Bcl-2 and inhibition of Bax expression [48], it might be imagined that in this experiment punicalagin inhibited JAK/STAT pathway followed by the reversed expression profile of Bcl-2/Bax, which eventually induced the process of apoptosis.

### 2.7. Punicalagin against Osteosarcoma

Huang et al. [30] demonstrated Pug to induce apoptosis in three human osteosarcoma cell lines (U2OS, MG63, and SaOS2); 48-h treatment with 100 μM Pug led to the substantial increase in early and late apoptotic cells in all three cell lines. Furthermore, 24-h Pug treatment of the three cell lines reduced the cells invasiveness in Matrigel assay. Additionally, the authors investigated Pug effect on the NF-κB signaling route and found the inhibition of active (phosphorylated) form of IκBα as well as the prevention of NF-κB-p65 translocation to the nucleus in two cell lines (U2OS and SaOS2). However, no impact on mTOR signaling has been noted. Moreover, antiangiogenic and anticancer effects of Pug have been observed by the authors on xenografted mice model, where the injection of Pug attenuated the growth of osteosarcoma, as well as reduced the tumor neoangiogenesis.

### 2.8. Punicalagin against Glioma

Wang et al. [37] demonstrated that 24- and 48-h Pug-treatment of human U87MG glioma cells significantly decreased cell viability in a dose-dependent manner (in the concentrations up to 30 μg/mL), arresting cell cycle at G2/M phase (reflected by Pug downregulation of cyclins A and B). Additionally, the authors analyzed Pug impact on chosen factors associated with the stimulation of cell lethality through two mechanisms, namely apoptosis and autophagy. Their findings led them to conclusions considering the involvement of both pathways in Pug-stimulated attenuation of cancer progression. Namely, Pug induced caspases 3 and 9 levels, activated PARP cleavage, as well as decreased Bcl-2 expression, all being characteristic apoptotic features. On the other hand, the application of caspases inhibitor only partly reversed Pug lethal effect exerted on the cells, which indicated the participation of an alternative route. Therefore, the authors checked for the autophagy markers, and noted the Pug-induced accumulation of microtubule-associated protein light chain 3 II (LC3-II), as well as punicalagin-stimulated phosphorylation of AMP activated kinase (AMPK) and p27^Kip1^. Therefore, the authors suggested the Pug-induction of autophagy through the upregulation of LKB1-AMPK pathway.

## 3. Punicalagin-Attenuated Signaling Pathways with Reference to Hallmarks of Cancer

### 3.1. Hallmarks of Cancer

Despite an enormous complexity and diversity of cancer diseases, some common characteristics attributable to most of the neoplasia types have been formulated. These hallmarks of cancer shaped by Hanahan and Weinberg [49,50] include resisting cell death, sustained proliferative signaling, activating invasion and metastasis, deregulating cellular energetics, enabling replicative immortality, inducing angiogenesis, avoiding immune destruction, and evading growth suppressors. Genome instability increasing mutations rate, as well as tumor promoting inflammation aid in the acquirement of these traits. This concept has been recently elaborated by Senga and Grose [51], who added four more features characterizing cancer pathology: dedifferentiation/transdifferentiation, epigenetic dysregulation, altered microbiome, and altered neuronal signaling. Therefore, for better comprehension of neoplastic disorder, one should analyze it in context with the local environment of the host tissue, as well as the whole organism. Punicalagin interference with signaling circuits disturbed in neoplasias, outlined earlier, is further addressed in the ensuing paragraphs with reference to the modified cancer hallmarks.

#### 3.1.1. Resisting Cell Death/Evading Growth Suppressors

The most characteristic feature of cancer cells is the ability to evade death and growth suppression. To this aim, pathological cells alter their metabolism to avoid apoptosis and take advantage of autophagy machinery to survive in unfavorable conditions. In normal cells, apoptosis (programmed cell death) is inevitable for the removal of destroyed cells to maintain tissues in healthy condition. Therefore, damaged cells undergo shrinking, membrane blebbing, and can be removed by macrophages without spilling their contents into the environment and stimulating inflammatory processes (like in the case of necrotic death). Apoptosis can be induced in the aftermath of internal or external events. The intrinsic (mitochondrial) pathway is initiated by receptor-independent signals generated in the cell, such as irradiation or reactive oxygen species accumulation. The major regulator responding to the level of DNA damage and further determining cell fates is the tumor suppressor p53 protein, which in turn controls Bcl-2 family proteins, from which Bcl-2 and Bcl-XL belong to pro-survival, whereas Bax or Bak stimulate apoptotic events. With a strong proapoptotic signal (e.g., when the accumulation of genetic errors seems to be unrepairable), they increase the permeability of the inner mitochondrial membrane and release of proapoptotic factors, including cytochrome C. Extrinsic pathways are initiated by the induction of multiple death receptors which activate caspase cascade starting from caspase 8. Both pathways converge on the execution phase initiated by the activation of caspase 3 [52]. Since cancer cells gain the capabilities of circumventing apoptotic death, chemotherapy as well as application of natural phytochemicals have been focused on the attempts to counteract this phenomenon. Hence, as discussed earlier, Pug proapoptotic actions deliver hope in reversing the malignant phenotype. Punicalagin was shown to induce apoptosis via the upregulation of caspases and Bax and/or induction of cytochrome C release from the mitochondrium, as well as downregulation of Bcl-2/Bcl-XL in human cervical [19,20], ovarian [27], lung [28,29], and colorectal [33] cancer cell lines.

Autophagy (“self-eating”) is a conservative process employed by normal cells to recycle the wastes (destroyed/abnormal intracellular structures/macromolecules) and reuse their building blocks (such as amino acids) for the synthesis of new proteins and other cellular components, as well as for the generation of energy. Although basal autophagy proceeds constitutively at a low level, it heightens with nutrient/oxygen deprivation to increase the survival odds under stressful conditions. Autophagy can be further divided into macroautophagy, selective autophagy, microautophagy, and chaperone-mediated autophagy, from which macroautophagy, the most common and best studied type, is usually referred to as simply autophagy [53]. Macroautophagy proceeds through the formation of an autophagosome (a sequestration of a fragment of cytoplasm by surrounding its contents with double membranes) and then its fusion with lysosome (yielding autophagolysosome) to deliver hydrolases required for the degradation of the enclosed molecules [54]. This process requires five stages: induction, nucleation, autophagosome extension, maturation, and autophagolysis [55,56]. Signaling pathways regulating those events include mammalian target of rapamycin (mTOR), as well as AMP-activated protein kinase (AMPK) routes. mTOR being active at sufficient nutrient satiation suppresses the induction of autophagy, whereas during starvation, inflowing signals block mTOR route, which triggers autodigestion mechanism. In turn, AMPK being the sensor of intracellular energy level, gets activated at ATP depletion, therefore it induces autophagy at energy exhaustion. An array of factors need to be recruited to form complexes enabling the transition via all of the stages for the successful completion of autophagy. For instance, in the induction stage the formation of the following complex: ULK-Atg13-FIP-200-Atg101 is required, whereas in the further steps, except for other proteins, beclin1, p62/SQSTM1 (sequestosome) and LC3 (microtubule-associated protein light chain 3) system are inevitable. When it comes to autophagy’s role in cancer disease development, it is equivocal, often referred to as a “double-edged sword”. On one hand, properly functioning cell autodigestion protects from cancerogenesis, removing genetically unstable, damaged materials, thus maintaining tissue homeostasis. Therefore, some autophagy-related factors seem to work as tumor suppressors in the incipient neoplasias. An example is beclin-1 (the protein indispensable in the membrane nucleation—phagophore formation stage, as well as autophagosome maturation [55]) being attenuated in some cancers, such as breast [57,58], ovarian [57,59], prostate [57], cervical [60], lung [61], liver [62,63] cancer, osteosarcoma [64], and glioblastoma [65]. However, after the initiation of cancer development, the altered phenotype seems to employ autophagic machinery to survive in unfavorable conditions of the host microenvironment, e.g., under stressful circumstances associated with host immune system response, or starvation and oxygen deprivation, before the formation of new blood vessels after metastasizing to a distant locus. For this reason, cancer cells upregulate autophagy, as might be referred from some studies. For instance, in colorectal cancer an overexpression of beclin-1 has been observed [66]. Nevertheless, the results are contradictory. For example, whereas Shen et al. [59] observed decreased beclin-1 expression in ovarian cancer, especially in more advanced stages, Cai et al. [67] noted its overexpression correlated positively with the better survival rates. Recent literature data describing autophagy function in cancer development, as well as signal transduction pathways implicated in this process, focused on anticancer therapy (including the assessment of phytotherapeutics) have been addressed in several review papers [53,55,56,68,69,70].

Both apoptosis and autophagy seem to have been responsible for the aforementioned punicalagin induction of human colorectal [31] and glioma [37] cancer cells’ death. Additionally, in the first case, the authors demonstrated the involvement of annexin A1 signaling suppressed by Pug, whereas in glioma cancer cells Pug treatment led to cell death through the activation of AMPK route. Punicalagin also stimulated the death of human thyroid cancer cells, although via different mechanisms from apoptosis. Cheng et al. [34] observed Pug-stimulated cell death through the activation of autophagy via the upregulation of MAPK pathway and inhibition of mTOR signaling. Additionally, Pug-induced cell death was shown to be associated with ATM-mediated DNA damage response [35], as well as the triggering of senescent phenotype in the same thyroid cancer cells [36]. Since the discovery that oncogene overactivation is able to paradoxically stimulate cancer cell senescence [71], this mechanism has gained attention as a possible chemopreventive target. The involvement of polyphenols in the initiation of senescence phenotype in cancer cells has been recently reviewed by Bian et al. [72]. Multiple signaling pathways triggered by stressors seem to be involved in the senescence induction, including DDR, p53 [72], as well as signaling associated with inflammatory phenotype via the Pug-induced up-regulation of NF-κB and interleukines, as shown by Cheng et al. [36].

As discussed earlier, punicalagin has been demonstrated to inhibit viability and proliferation, simultaneously enhancing neoplastic cell death via different mechanisms. The results of some studies are ambiguous, and hence challenging in terms of interpretation, due to the fact that many signaling pathways are interdependent and/or overlap employing the same signaling events/factors. Therefore, in some instances, it is difficult to clarify the exact machinery yielding lethal effects. Complex interrelationships occur also in the case of apoptosis and autophagy, from which both seem to employ such factors as p53, Bcl-2/Beclin, p62, or caspases (as discussed by Buzun et al. [53]). In particular, p53 protein, termed “the guardian of the genome”, is implicated in various routes, except for apoptosis or autophagy, depending on the physio/pathological context, after the cell cycle arrest, it can stimulate also the processes leading to senescence [73]. Additionally, the role of NF-κB pathway is multifaceted (as mentioned in Section 2.1). In some studies NF-κB attenuation by punicalagin has been associated with the induction of cells’ death through apoptosis [19,30], whereas in other experiments it was Pug-upregulation of NF-κB that led to cancer cell growth arrest via senescence phenotype induction [36]. Pug-modulated components involved in signaling pathways participating in apoptosis/autophagy induction are illustrated in Figure 2.

#### 3.1.2. Enabling Replicative Immortality/Dedifferentiation

Cancer cells employ the machinery observed during embryogenesis to undergo dedifferentiation and gain the potential of replicative immortality. To acquire a stem-cell-like phenotype, neoplastic cells take advantage of two main pathways: Hippo and Wnt evolutionary conserved signaling circuits. The Wnt pathway orchestrates stem cell pluripotency and defines cell fate during development via the regulation of cell differentiation, proliferation, adhesion/migration and polarity. Canonical Wnt signaling proceeds through β-catenin mobilization and translocation to the nucleus, followed by the induction of a variety of target gene expressions, including Myc, Cyclin D1, MMP-7. Non-canonical (β-catenin-independent) Wnt routes comprise planar cell polarity (PCP) signaling and calcium-dependent signal cascade. Gene expression abnormalities resulting from mutations and/or aberrant epigenetic modulation enable neoplastic cells to convert Wnt/β-catenin signaling pathway for their advantage to promote cancer progression. Wnt impaired functioning has been implicated in a variety of cancers including gynecologic malignancies such as human papillomavirus (HPV)-related cervical cancer and ovarian cancer, reviewed recently by McMellen et al. [74]. The role of Wnt/β-catenin pathway abnormality in cervical cancer (CC) has been lately discussed by Yang et al. [75]. Most of the research findings have pointed to its upregulation in CC (evidenced by the overexpression of Wnt ligands or underexpression of its inhibitory factors, such as active glycogen synthase kinase 3 β—GSK3β) [75]. For example, Ramachandran et al. [76] demonstrated a downregulation of Wnt inhibitory factor 1 (WIF1—a secreted Wnt antagonist) in all examined CCs models, whereas WIF1 re-expression restrained cancer cell proliferation by the induction of cell cycle arrest and apoptosis. The suppressive effect of WIF1 was accompanied by the reversal of the pathway’s gene targets expression, e.g., it lowered the expression of c-Myc and cyclin D1, in such a way leading to cell proliferation dampening. In addition, it downregulated Bcl-2 expression, thus stimulating apoptosis. A similar impact on Wnt/β-catenin pathway in cervical cancer cells has been observed in the aforementioned studies [20], where a downregulation of β-catenin, c-Myc, cyclin D1, as well as Bcl-2 was achieved by Pug treatment. Substantial scientific evidence also supports the hyperactivation of the Wnt/β-catenin pathway in ovarian cancer [74,77,78,79,80], where the overexpression of multiple components involved in this route has been reported. For example, Wnt ligands (LRP6—low-density lipoprotein receptor-related protein 6, and DVL3—segment polarity protein disheveled homolog) have been demonstrated to undergo amplification. Additionally, the mechanisms protecting β-catenin from degradation might be mobilized. These include aberrant activation of PI3K which in turn inhibits GSK3β via phosphorylation, thus preventing β-catenin degradation, or β-catenin gene (*CTNNB1*) mutation yielding β-catenin protein resistant to degradation [74]. Tang et al. [27] noted the Pug-induced inhibition of Wnt/β-catenin pathway in ovarian cancer cells reflected by the downregulation of β-catenin, as well as its targets: survivin and cyclin D1. Additionally, the authors, in their investigations on cervical and ovarian cancer cells, observed a Pug-induced reduction in matrix metalloproteinases (MMP2 and 9) paralleled by a rise in their inhibitors (TIMP2 and 3). These effects resulted in the attenuation of the neoplastic cells growth, proliferation, and migration. It might be supposed that Pug impact inhibited the dedifferentiation of the cells, preventing them from acquirement of the stem-like phenotype. Since Wnt/β-catenin pathway seems to play a major role in epithelial to mesenchymal transition (EMT) in many tumors including gynecological malignancies [74,78,81,82,83], it might be hypothesized that Pug effects are also associated with blunting EMT in the studied ovarian and cervical cancer cells. In both processes, the overexpression of pro-proliferative and pro-invasive factors is observed, including cyclin D1 and metalloproteinases downregulated by Pug in Tang et al. [20,27] experiments. Due to its pivotal role in the development of gynecological malignancies, Wnt/β-catenin circuitry nodes have been tried as promising therapeutic targets (discussed by McMellen et al. [74]). Therefore, the aforementioned preliminary studies on Pug silencing impact on this pathway in cervical and ovarian cancers seem to be a promising approach. More direct evidence of the inhibitory impact of Pug on EMT has been demonstrated by Pan et al. [26] in breast cancer cells and is considered further in the subsequent paragraph.

#### 3.1.3. Activating Invasion and Metastasis/Transdifferentiation

Cancer cells switch regular cellular mechanisms to achieve the capabilities of invasion (expansion into adjacent host environment) and metastasis (migration to a new site and initiation a new tumor lesion). One of the machineries employed is a circuitry associated with adherens junctions signaling. Adherens junctions are dynamic structures involved in cell-cell, and cell-ECM interactions, which play an important role in the organization of tissues and organs. Abnormally activated signal-transduction pathways enable pathological cells to reshape their phenotype, in the aftermath of which they lose the attachment with the environment and degrade ECM components, acquiring the capability to translocate. Multiple tumors, including breast [84,85,86] and ovarian [81] cancers, increase their motility taking advantage of epithelial to mesenchymal transition (EMT) program. This process is connected with the transformation of “stationary” epithelial cells into “mobile” mesenchymal cells (passing through an array of intermediary phases [87]). It seems to be the major driving force which enables neoplastic cells to release all of the bonds with the adjacent cells and extracellular matrix, evade anoikis (cell death induced by attachment loss), and metastasize. Multiple factors and pathways are involved in EMT regulation, for example signaling associated with Wnt/β-catenin, transforming growth factor β (TGF-β), receptor tyrosine kinases (RTKs), signal transducer and activator of transcription 3 (STAT3), ECM-mediated routes, hypoxia-initiated chain events, etc. Among the features indicative of EMT is the downregulation of E-cadherin (being characteristic for epithelial cells), associated with the upregulation of N-cadherin (typical for mesenchymal phenotype) as well as the increase of matrix metalloproteinases (MMP2 and 9) [88]. This expression pattern was shown to be reversed by Pug treatment in human breast adenocarcinoma and resulted in the inhibition of the cells’ mobility, suggesting the suppression of EMT [26], which makes Pug a hopeful factor being able to attenuate invasive/metastatic potential of breast cancer cells through EMT inhibition. As discussed earlier, a similar mechanism of Pug impact might be ascribed to its action observed in other gynecological cancers, where Pug, via the inhibitory influence on Wnt/β-catenin pathway, reversed the expression pattern of metalloproteinases and their inhibitors participating in ECM modelling. Through down-regulation of MMP2 and 9 paralleled by up-regulation of their inhibitors (TIMP2 and 3), Pug attenuated the migratory potential of human cervical and ovarian cancers [20,27]. Therapeutic approaches in cancer treatment focused on EMT program limitation, with respect to medicinal drug repurposing, have been summarized and discussed by Ramesh et al. [89] and Voon et al. [90]. Pug-modulated components involved in signaling pathways participating in cancer progression are illustrated in Figure 2.

## 4. Punicalagin Application in Anti-Cancer Therapy?

While discussing the putative usage of punicalagin as an antineoplastic agent, the following issues require further corroboration: the actual active substance(s) which exert anticancer functionality ascribed to Pug should be defined, and the compromised feasibility of extrapolated ex-vivo study findings into in-vivo systems must be considered.

### 4.1. Which Substance Actually Works?

Some studies point to mixtures of phytochemicals which exhibit higher activity in comparison with purified compounds. Such a phenomenon has been observed by Seeram et al. [18] and Adams et al. [32], who noted a greater impact of pomegranate juice or pomegranate tannin extract on the studied cancer cell lines, in comparison with pure punicalagin. Therefore, they suggested synergistic actions of other biologically active components present in the mixtures, which reinforce their anticancer impact. Reversely, there are findings ascribing greater antineoplastic potential to Pug products released during spontaneous hydrolysis, such as ellagic acid, as was found by Larrosa et al. [33]. Their in-depth analysis indicated that it was rather ellagic acid which induced apoptosis in Pug-treated human colon adenocarcinoma.

### 4.2. What Happens after Punicalagin Ingestion?

Since most of the research investigating the impact of Pug on cancer has been performed in in vitro cell line models, there is still a long way to go before translating the obtained results into an in vivo system. While considering the application of punicalagin in anticancer therapy, its processing in the human organism should be addressed. Firstly, when taken orally, punicalagin passes the subsequent segments of the gastrointestinal tract, being subjected to low pH in the stomach, as well as digestive hydrolases. Secondly, in the large intestine, it is acted upon by microorganisms forming the host specific microbiota. Thirdly, in the intestinal and liver cells, Pug’s products are processed by phase 2 biotransformation reactions associated with the attachment of a variety of molecules, such as glucuronic or sulphuric acids. Therefore, there will be different Pug-associated compounds acting on cancers residing in the gastrointestinal tract (e.g., colorectal cancer), and on cancers localized in various internal organs (like gynecological ones). The situation is even more complicated when central nervous system cancers, such as gliomas developing in the brain, are considered, due to the necessity to cross the blood–brain barrier.

#### 4.2.1. Ellagitannins Metabolism by Microbiota

As mentioned before, punicalagin belongs to ellagitannins, forming beside gallotannins a subgroup of hydrolyzable tannins. Ellagitannins can be distinguished from gallotannins by the presence of a specific hexahydroxydiphenyl group or its derivatives, such as valoneoyl, sanguisorboyl or gallagyl residues. Hydrolyzable tannins are readily water soluble, nevertheless their high molecular weight (usually above 500 Da) hinders their efficient transport across biological membranes. Furthermore, their structural components such as aromatic rings, as well as phenolic and carbonyl groups, can participate in interactions with amino acids of proteins and peptides. It is these interactions that underlie their astringent and obstructive properties connected with the ability to bind to the intestinal epithelium components and exert a modulatory impact on membranous enzyme and receptor functions. The biotransformation of hydrolyzable tannins regardless of chemical structure basically occurs in the large intestine under the influence of microbiota, leading to the release of depsides and phenolic acids, such as gallic acid, ellagic acid, valoneoic acid lactone, sanguisorbic acid lactone, terminalin, etc. The resulting depsides belong to poorly water-soluble compounds with limited bioavailability. However, they provide nourishment for some microorganisms which convert them into even smaller structures, including urolithins and isourolithins (hydroxylated derivatives of dibenzo[b,d]-pyran-6-one [91]). The gut microbiota has been shown to be a critical factor influencing the bioactivity of ellagitannins. There is a well-known bidirectional interaction between tannins, their hydrolytic degradation products, and the microorganisms inhabiting the gut. Ellagitannins and released depsides such as EA can modulate the microbial population (prebiotic effect), and the gut microbiota can convert these large structures to metabolites that are more bioavailable than precursor compounds. The metabolism of hydrolyzable tannins by the gut microorganisms is not the same in all individuals, resulting in different metabotypes. In the studies evaluating the biotransformation of ellagitannins in humans, it has been noted that subjects can be divided into those who produce EA metabolites (producers) and those who do not (non-producers) [92]. In the work of Selma et al. [93], approximately 10% of the population (aged 5–90 years) did not produce urolithins. Three human urolithin-associated metabotypes have been described, i.e., metabotype 0 (not producing urolithin), metabotype A (producing urolithin A as a unique final metabolite), and metabotype B (generating urolithin B and/or isourolithin A, beside urolithin A). Although the production of some intermediate urolithins has recently been attributed to species of the Eggerthellaceae family named *Gordonibacter urolithinfaciens* and *G. pamelaeae*, the identification of the microorganisms responsible for the complete conversion of EA into final urolithins, especially those associated with B metabotype, is still incomplete [92]. *G. urolithinfaciens* and *G. pamelaeae* (the bacterial species engaged in the production of urolithins from ellagic acid), have been demonstrated to generate urolithins M5 (3,4,8,9,10-pentahydroxyurolithin), M6 (3,8,9,10-tetrahydroxyurolithin) and C (3,8,9-trihydroxyurolithin). They correlated positively with metabotype A, but negatively with metabotype B. A recently discovered *Ellagibacter isourolithinifaciens* was shown to be able to produce isourolithin A (3,9-dihydroxyurolithin) and positively correlated with metabotype B [94,95], whereas *Bifidobacterium pseudocatenulatum* INIA P815 strain was capable of producing urolithins A and B from ellagic acid [96].

#### 4.2.2. Urolithins Phase II Biotransformation

Urolithins formed in the colon are further subject to absorption and phase II biotransformation reactions, yielding their conjugated derivatives, mainly glucuronides and sulfates which have been identified in the urine and blood plasma of the studied individuals after ellagitannins intake. The metabolites detected in those body fluids after consumption of pomegranate fruit preparations include urolithin A, isourolithin A, urolithin B, but mainly their glucuronides or sulphates [97,98,99] which are most likely responsible for the antineoplastic activity of *P. granatum* in the internal organ cancers. In the gut, those metabolites may exist alongside the precursor structures of ellagitannins, further complementing their impact on e.g., colorectal cancer. Seeram et al. [99] analyzed metabolites of pomegranate components in body fluids of healthy volunteers who were administered pomegranate juice concentrate containing 318 mg punicalagin and 12 mg EA. After absorption of EA and urolithins, their phase II metabolites—methoxylated and conjugated with glucuronic and sulphuric acids—were detected in blood plasma and urine. Plasma EA concentration increased to 60 nM and returned to the baseline within 5 h. Additionally, urolithin A and B glucuronides appeared in blood plasma (at concentrations ranging from 20 to 110 nM) within 6 h after pomegranate juice concentrate intake. Moreover, in urine of some individuals, EA and urolithin derivatives were detected, namely dimethylellagic acid glucuronide (DMEAG—detected only on the day of ingestion) as well as urolithin A (in 16 of 18 participants) and B (in 5 of 18 participants) glucuronides. Urinary metabolites appeared after 12 h, in accordance with the rhythm of their formation by intestinal bacteria and the enterohepatic circulation of EA.

#### 4.2.3. Punicalagin Biotransformation

Like other ellagitannins, punicalagin follows analogical pattern of transformations in the human organism. It undergoes hydrolytic degradation, products of which are next acted upon by microbiota, yielding dibenzopyranone-type urolithins and isourolithins, as well as other low-molecular weight phenols formed from the acyl residues (hexahydroxydiphenyl and gallagyl groups) [100] (Figure 1). Compared to Pug, these released phenols lack astringent properties but retain a number of interesting activities beneficial to health, including anti-cancer properties. The newest data associated with chemopreventive functions of ellagitannin derivatives have been summarized by Al-Harbi et al. [101]. The authors discussed the antineoplastic impact of various urolithin types on prostate, breast, uterine, liver, colon, and bladder cancers, with respect to the modified signaling pathways. For instance, urolithin A has demonstrated antineoplastic properties against colon cancer through the inhibition of cell proliferation by reducing the glycolytic pathway [102], inducing autophagy [103], senescence [104], or apoptosis [105,106], and its activity was greater in comparison with urolithins C, D and B [104,107]. Moreover, urolithin A has been shown to compromise leukemic cell proliferation [108], lung cancer cell invasiveness (via EMT inhibition) [109], as well as pancreatic cancer growth (via PI3K/AKT/mTOR pathway attenuation) [110]. However, as observed by Gonzalez-Sarrias et al. [107] and Gimenez-Bastida et al. [104], urolithins glucuronidation dampened antiproliferative effect.

Notwithstanding the aforementioned degradation products of punicalagin, a specific ellagitannin product has been detected, namely a depside called terminalin [16] formed from the gallagyl group, but its further fate in the human body is unknown.

## 5. Concluding Remarks

In light of the presented data, punicalagin seems to be a promising chemopreventive measure in gynecological cancers, as well as thyroid, lung, colorectal, and other cancer types, regardless of the actual compound(s) which exert their beneficial effects. Punicalagin both in mixtures with other components of pomegranate preparations, as well as after hydrolysis and conversion into ellagic acid, urolithins, and other products, seems to protect against cancer development. Punicalagin per se has been demonstrated to affect multiple signaling pathways whose dampening reduced cancer cell viability and proliferation via the induction of cell death, employing apoptotic, autophagic, or senescent machineries. Additionally, punicalagin compromised the invasive potential of cancer cells through the attenuation of β-catenin pathway or down-regulation of GOLPH3 associated with epithelial to mesenchymal transition reduction. Migratory capacity blunting by punicalagin was also reflected by the down-regulation of matrix metalloproteinases and upregulation of their inhibitors. However, since most of the presented results derive from ex-vivo studies, they require in-depth consideration before being translated into practical application. For instance, to evade punicalagin metabolism after oral intake, and hence increase its bioavailability, other administration routes should be designed with the application of state-of-the-art approaches, such as nanotechnology, encapsulation, and specific targeting [111]. An encouraging antineoplastic effect of nanocapsule-enclosed punicalagin on breast cancer cells has been observed by Shirode et al. [112]. At all events, more scientific investigations are necessary to untangle the specific role of punicalagin in malignant, hijacked signaling pathways. Finally, it should be underlined that most of the scientific data presented and discussed here originate from in vitro cancer cell line models. Therefore, although the results are very promising, they cannot be directly extrapolated into the practical application in the human organism and need confirmation in in vivo models.

## Figures and Tables

**Figure 1 nutrients-13-02733-f001:**
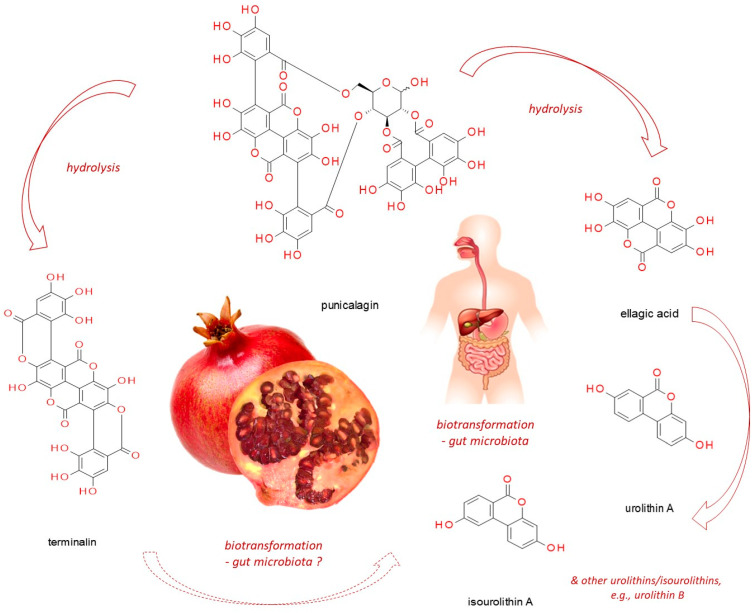
The structure of punicalagin and its derivatives (pomegranate photograph has been created by Izabela Fecka).

**Figure 2 nutrients-13-02733-f002:**
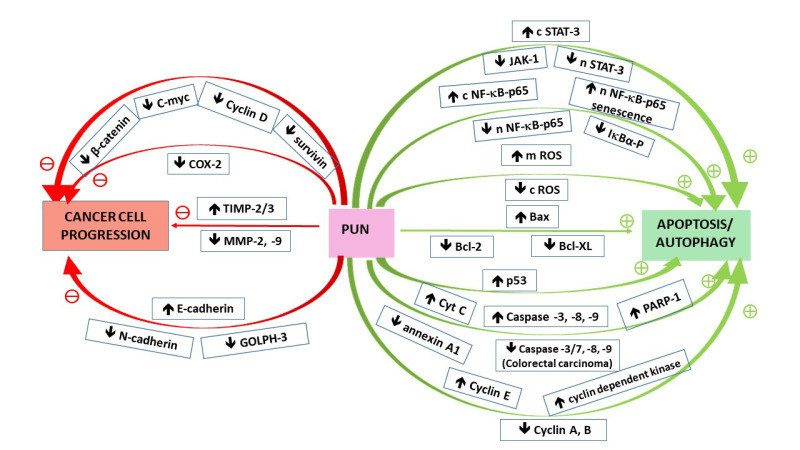
Effect of punicalagin on signaling pathways components associated with cancer cells development. The red and green arrows indicate the effect of particular compounds on either cancer cells progression or apoptosis/autophagy. Upwards and downwards arrows beside a given compound indicate positive or negative impact of a given compound on either cancer cells progression or apoptosis/autophagy. Abbreviations: GOLPH-3 = golgi phosphoprotein 3; MMP2/9 = matrix metalloproteinases 2 and 9; PARP(−1) = poly(ADP-ribose) polymerase (1); TIMP2/3 = tissue inhibitors of metalloproteinases 2 and 3; letters “c”, “n”, and “m” denote subcellular localization; cytosolic, nuclear, and mitochondrial, respectively.

**Table 1 nutrients-13-02733-t001:** Modulatory punicalagin impact on signaling pathways in human cancer cell lines.

Cancer Type	Experimental Model	Up-Regulation ↑	Down-Regulation ↓	Final Effect	Ref
Cervical carcinoma	ME-180 cervical cancer cell line	Bax; Casp-3 and 9; p53; cytosolic NF-κB-p65	Bcl-2; nuclear NF-κB-p65	Inhibition of cell proliferation and induction of apoptosis via suppressing NF-κB signaling	[19]
Cervix epitheloid carcinoma	HeLa cervical cancer cell line	Bax; TIMP2/3	Bcl-2; β-catenin, C-myc, cyclin D1; MMP2/9	Inhibition of cell proliferation and migration Cell cycle arrest at G1 phase Suppression of β-catenin pathway Induction of apoptosis	[20]
Breast adenocarcinoma	MCF-7 and MDA-MB-231 cell lines	E-cadherin	GOLPH3; MMP2/9; N-cadherin	Suppression of cell viability, EMT and migration via the regulation of GOLPH3	[26]
Ovarian cancer	A2780 ovarian cancer cells	Bax; TIMP2/3	Bcl-2; β-catenin, cyclin D1, survivin; MMP2/9	Inhibition of cell viability and migration Cell cycle arrest at G1 phase Suppression of β-catenin pathway Induction of apoptosis	[27]
Lung carcinoma	Lung cancer A549 cell line	Bax; Casp-3 and 9; cytochrome C; ROS; cytosolic STAT-3	Bcl-2; Jak-1; nuclear STAT-3	Inhibition of cell proliferation and induction of apoptosis via suppressing STAT-3 activation	[28]
Lung carcinoma	Lung cancer A549 cell line	Casp-3, 8, and 9; PARP-1; mitochondrial ROS	Cytosolic ROS	Induction of apoptosis; cell cycle arrest at G1/S	[29]
Osteosarcoma	Cell lines U2OS, SaOS2	-	Phosphorylated IκBα; nuclear NF-κB-p65; IL-6; IL-8	Inhibition of cell proliferation and induction of apoptosis possibly via suppressing NF-κB signaling Reduction of invasion potential	[30]
Colorectal carcinoma	Cell line HCT116	Cytochrome C	Annexin A1; caspases 3/7, 8 and 9	Induction of cell death via apoptosis and autophagy	[31]
Colon adenocarcinoma	Cell line HT-29	-	COX-2	Suppression of inflammatory cell signaling	[32]
Colon adenocarcinoma	Cell line Caco-2	Casp-3 and 9; Cytochrome C; Cyclin E	Bcl-XL; Cyclin A and B1	Cell cycle arrest at S phase; induction of apoptosis via the intrinsic-mitochondrial pathway stimulation	[33]
Papillary thyroid carcinoma	Cell line BCPAP	LC3-II conversion, beclin-1; phosphorylated ERK 1/2 and p38	p62; phosphorylated p70, S6, and 4E-BP1	Induction of apoptosis-independent cell death via autophagy through the activation of MAPK and inhibition of mTOR signaling	[34]
Papillary thyroid carcinoma	Cell line BCPAP	p-H2A.X; p-ATM	-	Induction of cells death via the ATM-mediated DNA damage response	[35]
Papillary thyroid carcinoma	Cell line BCPAP	SA-beta-Gal; cyclin-dependent kinase inhibitor p21; IκBα; nuclear NF-κB-p65; IL-6; IL-1β	-	Induction of senescent growth arrest and senescence-associated secretory phenotype (SASP) through the activation of NF-κB	[36]
Glioblastoma astrocytoma	Cell line U87MG	Casp-3 and 9; PARP; Cyclin E; LC3-II cleavage, AMPK-P, p27-P	Bcl-2; Cyclin A and B	Cell cycle arrest at G2/M phase; induction of cell death via apoptosis and autophagy	[37]

Abbreviations: AMPK-P = phosphorylated AMPK (AMP activated kinase); EMT = epithelial to mesenchymal transition; GOLPH3 = golgi phosphoprotein 3; LC3-II = microtubule-associated protein light chain 3 II; MMP2/9 = matrix metalloproteinases 2 and 9; PARP(−1) = poly(ADP-ribose) polymerase (1); p-H2A.X = phospohorylated histone 2A.X; p-ATM = phospohorylated ATM (ataxia telangiectasia mutated); p27-P = phosphorylated p27; SA-beta-Gal = senescence-associated beta-galactosidase; TIMP2/3 = tissue inhibitors of metalloproteinases 2 and 3.

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
