# Peer review of "Punicalagin in Cancer Prevention—Via Signaling Pathways Targeting"

_nutrients, 2021, doi:10.3390/nu13082733_

Round 1

Reviewer 1 Report

This is a review on the effects of Punicalagin in cancer prevention, with an emphasis on signaling pathways. Several article have been published on the pomegranate and its constituent compounds  and therefore the novelty is low (Syed et al., Anticancer Agents Med Chem. 2013 Oct; 13(8): 1149–1161; Turrini et al., Oxid Med Cell Longev. 2015; 2015: 938475; Venusova et al., Nutrients 2021, 13, 2150). A major issue with this polyphenol  is its low bioavailability and absorption. Therefore, most of the biological effects depicted are observed in vitro. The sections on Punicalagin in different types of cancer are mostly from in vitro studies, and therefore it is misleading to use them to depict the anticancer activity of the compound. 

Unfortunately a recent article on Punicalagin in cancer prevention has been published in Nutrients, and therefore I do not consider it sufficiently novel for Nutrients. 

Author Response

Dear Reviewer,

Thank you very much for your evaluation of the submitted paper.

Undoubtedly, a lot of scientific articles have addressed pomegranate effects in a variety of disorders. Several reviews have been published focusing on various aspects of punicalagin, among them the recent paper by Venusova et al. published in Nutrients and focusing on the effect of punicalagin on immune functions. However, much less is known about pure punicalagin with respect to cancer progression and there is no current update on recent findings in this area. Therefore, in our work we have tried to extract, summarize and discuss the data which originate from experiments on purified punicalagin in human cancer cell lines. As a result of the literature search, we have found 14 articles focusing directly on this topic, and most of them are pretty recent publications; 5 published in 2020, and one in 2021, 5 article appeared in the years 2016-2018. However, to complete the picture of punicalagin impact on signaling routes employed by neoplastic cells, we have also considered and summarized the results coming from older articles (one from 2013, and two from 2006 year).

Our intention has not, by all means, been to mislead the reader. Therefore, in the revised version of this manuscript, we have underlined that the discussed data originate mainly from in vitro cancer cell line models. This information has been added to the graphical abstract, entitled: “Punicalagin against cancer – from in vitro studies”, as well as placed in the regular abstract („Considerations presented in this review are based mainly on data obtained from in vitro cell line models…”).

Moreover, several other alterations have been introduced into the manuscript. Please, find them enlisted below.

  • The title of section 2 has been rephrased; from

“2. Punicalagin (Pug) in cancer prevention and therapy”

Into:

“2. Anti-cancer activities of punicalagin (Pug)”

Actually, in the previous form, this title might have put one on the wrong track, suggesting data extraction from observational or interventional studies.

  • The caption of table 1 has been rephrased from:

“Table 1. Modulatory punicalagin impact on signaling pathways in human cancer cells.”

Into:

“Table 1. Modulatory punicalagin impact on signaling pathways in human cancer cell lines.”

  • A question mark has been added to the title of section 4:
  1. Punicalagin application in anti-cancer therapy?
  • A short paragraph has been added to the „Final remarks” section, clarifying this issue („Finally, it should be underlined that most of the presented and discussed here scientific data originate from in vitro cancer cell line models. Therefore, although the results are very promising, they cannot be directly extrapolated into the practical application in the human organism, and need confirmation in in vivo”)

I hope that you find the above explanations and amendments to the manuscript satisfactory.

With best regards,

Izabela Berdowska

Reviewer 2 Report

The paper develops an interesting topic. English language should be revised. The conclusions should be extensively refined.  

Author Response

Dear Reviewer,

We truly appreciate your evaluation as well as comments on the submitted manuscript.

We have extensively revised the article according to your suggestions in the hope that its quality has improved. The conclusions have been refined; a paragraph has been added at the end, to summarize and close our considerations:

„Finally, it should be underlined that most of the presented and discussed here scientific data originate from in vitro cancer cell line models. Therefore, although the results are very promising, they cannot be directly extrapolated into the practical application in the human organism, and need confirmation in in vivo models.”

The manuscript has been corrected by a person proficient in English language. Therefore, the linguistic mistakes have been amended, as well as the style has been polished; many sentences have been rephrased to increase their clarity.

Since it might be misunderstood that the results discussed in our manuscript come from observational or interventional studies, we have underlined that the discussed data originate mainly from in vitro cancer cell line models also in other parts of the manuscript. This information has been added to the graphical abstract, entitled: “Punicalagin against cancer – from in vitro studies”, as well as placed in the regular abstract („Considerations presented in this review are based mainly on data obtained from in vitro cell line models…”).

Moreover, several other alterations have been introduced into the manuscript. Please, find them enlisted below.

  • The title of section 2 has been rephrased; from

“2. Punicalagin (Pug) in cancer prevention and therapy”

Into:

“2. Anti-cancer activities of punicalagin (Pug)”

  • The caption of table 1 has been rephrased from:

“Table 1. Modulatory punicalagin impact on signaling pathways in human cancer cells.”

Into:

“Table 1. Modulatory punicalagin impact on signaling pathways in human cancer cell lines.”

  • A question mark has been added to the title of section 4:

“4. Punicalagin application in anti-cancer therapy?”

Actually, in the whole section 4, there is a discussion emphasizing the long way between the assessment of punicalagin anti-cancer activities in cell line experiments (in vitro), and its practical application in anti-cancer therapy. We underline that after oral intake, punicalagin undergoes biotransformation reactions which convert it into a variety of derivatives including ellagic acid, as well as urolithins. Therefore, before its practical application, it must be more thorougly studied in in vivo models.

I remain in the hope that the above amendments/explanations sufficiently meet your requirements, and that in its upgraded form, our manuscript is eligible for publication.

With best regards,

Izabela Berdowska

Reviewer 3 Report

The review summarized the effect of punicalagin on cancer prevention. The review is well-written and structured. Section 4 is particularly interesting and gave an outlook on the following studies that will have to be performed to evaluate the in vivo effect of the compound. 

Minor comments: 

The credit for the photo of the pomegranate should be added in the caption of figure 1.

general: The name "authors" should not be capitalized

The verb "exhibited" should be in most of the cases replace by demonstrated

The verb "collected" should be in most of the cases replace by summarized

Line10: The sentence should be corrected to: " The extract of pomegranate (Punica granatum) is applied ..."

Line 13: Anticancer activities

line 14-15: a prevailing compound in pomegranate

line 20: Presented here are considerations

line 28: A reference needs to be added to the sentence

line 34: A reference needs to be added to the sentence

line 63: Remove the molecular formula, the name, and the CAS is sufficient

line 65: It is abundant in pomegranate juice

line 69-70: Remove the molecular formula; the name and the CAS is sufficient

line 71-73: the sentence should be rephrased

line 85: the references 19-22 did not support the claims

line 85: Remove "which seemed to enhance punicalagin impact". The authors hypothesized that other compounds increased the effect of punicalagin; however, no evidence supports this claim. It could also be another compounding the pomegranate juice that possesses a strong antiproliferative activity.

line 89: A reference needs to be added to the sentence "...cervical cell lines"

Author Response

Dear Reviewer,

We deeply appreciate your thorough evaluation as well as detailed comments on the submitted paper. We have corrected the article according to your suggestions in the hope that its quality has improved. Below, please find the meticulous references to your remarks:

Minor comments: 

The credit for the photo of the pomegranate should be added in the caption of figure 1.

The figure has been replaced by another one with pomegranate photo created by Izabela Fecka, what has been mentioned in the caption.

general: The name "authors" should not be capitalized - corrected

The verb "exhibited" should be in most of the cases replace by demonstrated - corrected

The verb "collected" should be in most of the cases replace by summarized - corrected

Line10: The sentence should be corrected to: " The extract of pomegranate (Punica granatum) is applied ..." - corrected

Line 13: Anticancer activities - corrected

line 14-15: a prevailing compound in pomegranate - corrected

line 20: Presented here are considerations

The whole sentence has been reformulated (lines 20 and 21)

line 34: A reference needs to be added to the sentence

The sentence has been rephrased and references have been added

line 63: Remove the molecular formula, the name, and the CAS is sufficient

The molecular formula has been removed

line 65: It is abundant in pomegranate juice

“the” has been removed from the sentence

line 69-70: Remove the molecular formula; the name and the CAS is sufficient

The molecular formulas have been removed and CAS numbers have been added.

line 71-73: the sentence should be rephrased

The sentences have been rephrased, from:

„Antimutagenic effects of both Pug and EA have been observed by Zahin et al. [17]. In their experiments testing different carcinogens, both compounds in a dose dependent manner significantly protected DNA from destruction.”

into:

„Both punicalagin and its derivative – ellagic acid have been demonstrated to protect DNA from mutations [17]. Zahin et al. [17] observed that these two compounds dose-dependently diminished the level of DNA damages caused by a variety of carcinogens.”

line 85: the references 19-22 did not support the claims

References 19-22 have been removed

line 85: Remove "which seemed to enhance punicalagin impact". The authors hypothesized that other compounds increased the effect of punicalagin; however, no evidence supports this claim. It could also be another compounding the pomegranate juice that possesses a strong antiproliferative activity.

The sentence has been rephrased.

line 89: A reference needs to be added to the sentence "...cervical cell lines"

The references have been added.

I remain in the hope that in the revised version the manuscript will be eligible for publication.

With best regards,

Izabela Berdowska